# The Hidden Burden of Severe Asthma: From Patient Perspective to New Opportunities for Clinicians

**DOI:** 10.3390/jcm9082397

**Published:** 2020-07-27

**Authors:** Nicola Scichilone, Peter John Barnes, Salvatore Battaglia, Alida Benfante, Robert Brown, Giorgio Walter Canonica, Gaetano Caramori, Mario Cazzola, Stefano Centanni, Antonella Cianferoni, Angelo Corsico, Giuseppe De Carlo, Fabiano Di Marco, Mina Gaga, Catherine Hawrylowicz, Enrico Heffler, Maria Gabriella Matera, Andrea Matucci, Pierluigi Paggiaro, Alberto Papi, Todor Popov, Paola Rogliani, Pierachille Santus, Paolo Solidoro, Alkis Togias, Louis-Philippe Boulet

**Affiliations:** 1Division of Respiratory Diseases, Department of Health Promotion Sciences, Maternal and Infant Care, Internal Medicine and Medical Specialties (PROMISE), University of Palermo, Piazza delle Cliniche 2, 90143 Palermo, Italy; salvatore.battaglia@unipa.it (S.B.); benfantealida@gmail.com (A.B.); 2Airway Disease Section, National Heart & Lung Institute, Imperial College London, Dovehouse Street, London SW3 6LY, UK; p.j.barnes@imperial.ac.uk; 3Department of Anesthesiology and Critical Care Medicine, Medicine, Department of Medicine, Division of Pulmonary Medicine, Department of Environmental Health and Engineering, Johns Hopkins University, Baltimore, MD 21287, USA; rbrown@jhmi.edu; 4Personalised Medicine Clinic Asthma & Allergy, Humanitas University, Department of Biomedical Sciences, IRCCS Humanitas Research Hospital, Rozzano, 20089 Milan, Italy; giorgio_walter.canonica@hunimed.eu (G.W.C.); enrico.heffler@hunimed.eu (E.H.); 5Respiratory Medicine Unit, Department of Biomedical Sciences, Dentistry and Morphological and Functional Imaging (BIOMORF), University of Messina, 98122 Messina, Italy; gcaramori@unime.it; 6Unit of Respiratory Medicine, Dept. Experimental Medicine, University of Rome “Tor Vergata”, 00133 Rome, Italy; mario.cazzola@uniroma2.it (M.C.); paola.rogliani@uniroma2.it (P.R.); 7Respiratory Unit, ASST Santi Paolo e Carlo, San Paolo Hospital, Department of Health Sciences, University of Milan, 20142 Milan, Italy; stefano.centanni@unimi.it; 8Pediatrics Department, Perlman School of Medicine, University of Pennsylvania, Philadelphia, PA 19104, USA; cianferonia@email.chop.edu; 9Division of Respiratory Diseases, IRCCS Policlinico San Matteo Foundation and Department of Internal Medicine and Therapeutics – University of Pavia, 27100 Pavia, Italy; corsico@unipv.it; 10The European Federation of Allergy and Airways Diseases Patients Associations (EFA), 1000 Brussels, Belgium; giuseppe.decarlo@efanet.org; 11Respiratory Unit, ASST - Papa Giovanni XXIII Hospital, Bergamo, University of Milan, 24127 Milan, Italy; fabiano.dimarco@unimi.it; 127th Respiratory Medicine Dept, Asthma Cen, Athens Chest Hospital, 11527 Athens, Greece; minagaga@yahoo.com; 13Division of Asthma, Allergy and Lung Biology, King’s College London, Guy’s Hospital, London SE1 9RT, UK; catherine.hawrylowicz@kcl.ac.uk; 14Unit of Pharmacology, Dept. Experimental Medicine, University of Campania “Luigi Vanvitelli”, 80138 Naples, Italy; mariagabriella.matera@unicampania.it; 15Immunoallergology Unit, Careggi University Hospital, 50139 Florence, Italy; andrea.matucci@unifi.it; 16Department of Surgery, Medicine, Molecular Biology and Critical Care, University of Pisa, 56126 Pisa, Italy; pierluigi.paggiaro@unipi.it; 17Research Center on Asthma and COPD, Dept of Medical Sciences, University of Ferrara, 44121 Ferrara, Italy; ppa@unife.it; 18Clinic of Occupational Diseases, University Hospital Sv. Ivan Rilski, 1431 Sofia, Bulgaria; ted.popov@gmail.com; 19Division of Respiratory Diseases, Department of Biomedical and Clinical Sciences (DIBIC), Università degli Studi di Milano, Ospedale L. Sacco, ASST Fatebenefratelli-Sacco, 20157 Milan, Italy; pierachille.santus@unimi.it; 20Pneumology Unit U, Cardiovascular and Thoracic Department, AOU Città della Salute e della Scienza di Torino, University of Turin, 10126 Turin, Italy; psolidoro@cittadellasalute.to.it; 21National Institute of Allergy and Infectious Diseases, Bethesda, MD 20814, USA; togiasa@niaid.nih.gov; 22Quebec Heart and Lung Institute, Laval University, Quebec City, QC G1V 0A6, Canada; lpboulet@med.ulaval.ca

**Keywords:** severe asthma, precision medicine, biomarkers, patient’s perspective

## Abstract

Severe asthma is an important topic in respiratory diseases, due to its high impact on morbidity and mortality as well as on health-care resources. The many challenges that still exist in the management of the most difficult-to-treat forms of the disease, and the acknowledgement of the existence of unexplored areas in the pathophysiological mechanisms and the therapeutic targets represent an opportunity to gather experts in the field with the immediate goals to summarize current understanding about the natural history of severe asthma and to identify gaps in knowledge and research opportunities, with the aim to contribute to improved medical care and health outcomes. This article is a consensus document from the “International Course on Severe Asthma” that took place in Palermo, Italy, on May 10–11, 2019. Emerging topics in severe asthma were addressed and discussed among experts, with special focus on patient’s needs and research opportunities, with the aim to highlight the unanswered questions in the diagnostic process and therapeutic approach.

## 1. Introduction

Severe asthma can be limiting for patients and frustrating for clinicians. Patients suffering from the most severe forms of the disease have the greatest impairment in their quality of life and generate the highest medical and societal costs. On the other hand, clinicians face difficult to treat symptoms and concomitant conditions that negatively influence the natural course of the disease. In this context, experts in the field gathered in Palermo, Italy, to participate in the “International Course on Severe Asthma” on May 10–11, 2019. Course participants represented expertise in asthma, pulmonology, allergy/immunology and emergency medicine and discussed emerging topics in severe asthma, with a special focus on unmet needs in education and research. The choice of the topics was based on a Delphi consensus that was conducted among general practitioners and pulmonologists in the area of Palermo in the year preceding the course, and those that were selected are summarized in Table 1. The urgency for the meeting came from the need of a consensus on the definition of severe asthma. In 1999, the European Respiratory Society (ERS) defined *difficult to treat* and *severe asthma* as “asthma, poorly controlled in terms of chronic symptoms, with episodic exacerbations, persistent and variable airway obstruction and continued requirement for short-acting beta-2-agonists and a reasonable dose of inhaled corticosteroids” [1]. Since then, the definition has evolved: in 2000, the American Thoracic Society (ATS) defined *refractory asthma* on the basis of two major criteria—1. treatment with oral corticosteroids > 50% of the time, or 2 high doses of inhaled corticosteroids (>1200 µg beclomethasone equivalent)—and at least two minor criteria, including requirement for daily treatment with short-acting beta-2-agonists (SABA), theophylline or leukotriene-receptor antagonist (LTRA), daily asthma symptoms requiring rescue medication, persistent airway obstruction; diurnal peak expiratory flow (PEF) variability > 20%, one or more urgent care visits for asthma per year, three or more oral steroid bursts per year, prompt deterioration with > 25% reduction in oral or inhaled corticosteroids (ICS) dosing and/or near fatal asthma event in the past [2]. This definition was applicable only when other conditions had been excluded, exacerbating factors optimally treated and poor adherence were not confounding issues. In 2006, GINA (Global INitiative for Asthma) proposed the first separation in the definition of *severity* and *control* and, in 2007, a consensus report included for the first time the element of “time” and of “specialist follow-up and treatment” [3]. This report stated that *severe asthma* applies to patients who have refractory asthma who remain difficult to control despite an extensive re-evaluation of diagnosis and management, and following an observational period of at least six months by an asthma specialist. The first ERS-ATS guidelines on severe asthma in adults and children were published in 2014 [4]. Severe asthma was defined as “asthma that requires treatment with high dose inhaled corticosteroids plus a second controller and/or systemic corticosteroids to prevent it from becoming “uncontrolled” or that remains “uncontrolled” despite this therapy”. Emphasis was placed on the necessity to confirm the diagnosis of asthma and exclude other conditions that may mimic asthma. In addition, the document recognized that severe asthma is a heterogeneous condition consisting of different phenotypes. Once the disease is properly characterized, the therapeutic targets can be identified, and proper treatment can be adopted (Table 2).

## 2. The Different Faces of Severe Asthma

### 2.1. The Patient’s Perspective 

The European Federation of Allergy and Airways Diseases Patients Associations has identified some of the challenges and needs for severe asthma patients to raise the awareness on the impact of severe asthma in patients’ and caregivers’ lives. First, to better control asthma, it is key to have a timely and accurate diagnosis. Many patients wait for years before knowing which subtype of asthma they suffer from and finally get a tailored and more effective treatment. Indeed, whereas an initial diagnosis of asthma is commonly posed in primary care settings, a formal diagnosis of severe asthma requires a more complex assessment following referral to a respiratory specialist. Although many advances have been made in diagnostic assessment, there is an urgent need to grant access to the best tools for all patients. Similarly, general practitioners need to be able to understand the results of the diagnostic tests and to refer to specialists when needed. Shortening the patient journey is key to improving the health-related quality of life for patients with severe asthma. In this respect, patients who present to their general practitioner with difficult-to-manage asthma should be adequately assessed using a structured and standardized methodology in order to avoid inappropriate escalation of treatment, and streamline clinical assessment and management, therefore optimizing patient referrals. Establishing clear referral pathways for patients with severe asthma is the first mandatory step to help patients receive early and appropriate treatment.

Besides diagnostics, treatments have also improved and new drugs, such as biologics, are changing the life of patients with severe asthma. Of course, the choice of the biologic drugs should also take into consideration the individual needs of the patient. Patients are not always aware of the long-term adverse effect of oral corticosteroids (OCS), and biologic drugs primarily aim at reducing, or abolishing, their chronic use. On this basis, there is a growing call for severe asthma care to be less reliant on the long-term use of OCS to prevent asthma attacks. The involvement of patients in research is key to improve clinical developments in respiratory disease, in phenotyping severe asthma and taking full advantage of personalized and precision medicine. Finally, the support of family, friends and especially other patients, can motivate patients in keeping adherence to the treatment. Parents or carers of young people with asthma need to know how to cope with the disease and their education is essential for a good care of the patient. In this scenario, communication is crucial. Patients should receive relevant information from their health-care professionals in a simple and clear format to better understand the treatment options and the consequences of different management approaches. The importance of good communication between health professionals and patients is recognized by the latter, and improving communication is strongly advocated for a satisfactory outcome. Patients ask for better skills to cope with their disease, and knowledge to properly recognize signs of asthma attacks or avoid unnecessary or ineffective practices. Constant education on correct inhaler technique is also important to ensure the optimal effect of currently prescribed medications. There should be shared decision-making between patients and their clinicians to ensure that optimal care is delivered. 

### 2.2. The Clinician’s Perspective

When approaching a patient suffering from the most severe forms of asthma, who has reasonably already seen several consultants, the clinician is forced to ask himself/herself the following questions: is it really asthma? Can I “cure” him/her? What does he/she expect from me? How do I manage exacerbations? How can I avoid further exacerbations? From a practical point of view, these questions translate into the following themes: diagnosis/differential diagnosis; personalized/precision medicine; prognosis; treatment of emergencies; barriers to health-care resources. The first question leads to a “dynamic” process, meaning that it needs to be relaunched especially when patients do not respond to current treatment. The diagnostic algorithm when approaching someone suffering from severe asthma resembles that of asthma in its milder forms, and incorporates differential diagnoses; indeed, confirmation of asthma requires that conditions that can mimic severe asthma (both pulmonary and non-pulmonary) are ruled out. Indeed, symptom misattribution can generate confusion in the diagnostic process. In this respect, objective measurements of variability in airway obstruction could contribute to elucidate the clinical picture, and should always be pursued. However, bronchial obstruction reversibility may be hard to achieve in severe asthmatics with long-standing disease, due to the remodeling changes of the airway wall, therefore moving to an area of diagnostic uncertainty. This process also includes the identification and treatment (or removal) of contributory factors, such as non-adherence, poor inhalation technique, triggers and comorbidities. Methods to assess non-adherence are mandatory in this population, and effective educational interventions aiming at avoiding intentional and non-intentional non-adherence should be encouraged in daily practice. These steps allow to move from a condition of difficult-to-treat asthma to that of severe asthma, which leads to the correct therapeutic approach. 

The second question raises from the shift paradigm in treating severe asthma, that is, anticipating the biologics prior to the administration of long-term use of systemic corticosteroids, and poses a challenge in terms of selection of the right treatment. The skills acquired through experience about the criteria needed to choose among the biologic drugs are essential and require continuous educational training and gaining of expertise. Eventually, what will guide therapy in severe asthma is not only efficacy, safety and cost, but the ability to modify the natural course of the disease, which answers the third question (prognosis). In this scenario, the respiratory community should strongly encourage head to head clinical trials and those specifically looking at the search for disease modifiers. The fourth question is extremely important: perhaps, learning about the exacerbation-prone phenotype in asthma could help to prevent severe exacerbations. In this regard, written action plans could help to recognize early signs of exacerbations, or worsening of symptoms, providing instructions for the management of acute conditions. Finally (fifth question), the clinician pretends to have the tools and facilities to properly treat his/her patients. In this regard, dedicated severe asthma service, or referral centers, have been demonstrated to improve asthma control and quality of life, and reduce exacerbations and emergency room visits, health-care use and oral steroid burden [6]. In this context, guidelines [5] advise that asthmatic patients be managed by an experienced specialist multidisciplinary team (MDT). An MDT case management approach has been shown to significantly reduce hospitalization in difficult asthma patients with prior frequent admissions [7]. The MDT can allow the identification of modifiable factors contributing to poor control, and proper management of overlapping conditions. Ideally, the severe asthma MDT should include asthma educators and specialists [8] such as otorhinolaryngologists, respiratory/allergy consultants and occasionally endocrinologists, rheumatologists, gynecologists, radiologists and immunologists. Other collaborators such as nutritionists, physiotherapists and psychologists may contribute to improve the outcomes by facilitating the communication with patients and application of treatment strategies (Figure 1). A coordinated multidisciplinary action can be particularly effective if expert primary care physicians are actively involved, both in the discussion of the actions to be taken, and in the opportunities for high-quality advanced training. Nowadays, severe asthma is no longer an orphan disease.

### 2.3. The Funding Agency’s and the Researcher’s Perspective 

From the perspective of a funding agency, the definition of severe asthma needs to be broad enough in order to avoid restricting research proposals and stifling the diversity of research approaches in the field. Research on severe asthma should involve patients who require high-dose ICS for asthma control, those who cannot be controlled even with high-dose treatment and those who experience frequent exacerbations. The latter group is particularly important when childhood asthma is considered [9]. Funding agencies such as the National Institute of Allergy and Infectious Diseases (NIAID), which is part of the US National Institutes of Health (NIH), support research projects that target various aspects of severe asthma, with focus on prevention and management. In the context of these targets, it is imperative that further understanding of the pathophysiology of severe asthma is attained. For NIAID, the focus for pathophysiology research is on the role of the immune system, infectious processes and airway homeostasis. Some principles that research on severe asthma needs to follow include careful and in-depth immunophenotyping in association with functional airway/lung phenotyping, consideration of both inherent and environmental inducers and their interaction, utilization of longitudinal, as opposed to cross-sectional, cohort studies and a combination of both agnostic, hypothesis-generating research and hypothesis-driven research within the same project. Longitudinal studies allow us to observe disease oscillations within each patient and obtain a clearer picture of factors and pathways that contribute to disease worsening (e.g., exacerbations) or disease stability. With the advent of -omics methodologies, where the entire universe of a research field can potentially be captured, hypothesis-generating research has stopped being a “fishing expedition” and has become essential in reducing bias and promoting discovery. In handling -omics data, it is important to combine multiple outcomes through a systems biology approach and to involve bioinformatics in designing studies and conducting innovative data analyses [10]. In severe asthma research, animal models can be used to test specific hypotheses related to the biologic role of a single molecule or a gene/molecular network identified through human research. However, we should be cautious of the reverse approach, i.e., animal models claiming to reflect severe asthma being used as the primary source of data for unveiling the pathophysiology of the disease.

Prevention of severe asthma is a worthy goal. However, if a specific preventive intervention for severe asthma were to be employed, we would need sensitive and specific early stage detection methodologies to identify individuals, most likely children, at risk. Such methodologies are not currently available. Furthermore, because severe asthma is a relatively rare condition, a large screening effort would be required to identify individuals at risk. On the other hand, because severe asthma is a sub-phenotype of a larger, persistent asthma phenotype, it is possible that a preventive intervention targeting a wider population may also impact the incidence of severe disease. For example, early multiple allergen sensitization in conjunction with recurrent wheezing constitutes a high-risk phenotype for the development of asthma [11]. A subgroup of this phenotype appears to develop early airway obstruction and may very well represent a specific group at risk of severe asthma. A preventive intervention could be applied to such an entire multi-allergic/early wheezing group, as even those children who may not end up developing severe asthma would benefit by not developing any chronic lower respiratory disease. 

Exacerbation-prone asthma should be considered part of the severe asthma spectrum. This is a phenotype particularly relevant to the pediatric population. The importance of preventing the development of this phenotype is that asthma exacerbations impact quality of life and education, incur costs, can lead to major acute respiratory ailments and may also result in reduced lung function over time [12]. Although asthma exacerbations are in their majority associated with upper respiratory viral infections, the pathophysiologic pathways through which a viral infection converts to an asthma exacerbation have only recently begun to be elucidated [12]. Furthermore, non-viral exacerbations are not infrequent and their cause remains unknown. In addressing these knowledge gaps, studies should involve longitudinal cohorts of individuals with exacerbation-prone asthma where upper and lower airway, as well as systemic outcomes, are captured at the beginning of any cold symptoms, a few days later or at the beginning of worsening asthma symptoms, regardless if those are preceded by upper respiratory symptoms [13]. In addition to careful documentation of clinical symptoms and airway (nasal, lung) function, these outcomes should include unbiased approaches such as nasal transcriptomics (possibly with single cell analysis), proteomics and upper airway virome and micro/mycobiome analysis. Concomitant sputum analyses would be ideal, but possibly limited by lower airway clinical status. Such studies would not be complete without the collection of environmental data to assess the relationship of asthma exacerbations with changes in the exposome. These studies require major resources, but, given their potential, funding agencies should be able to offer appropriate support.

### 2.4. The Scientific Societies’ Perspective 

Big data are nowadays providing a new scenario in evaluating several aspects of medicine. Relevant insights are coming from registries whose structures are collecting standardized data. As far as severe asthma is concerned, structured registries are currently available at regional, national and international levels, thus allowing an analysis at country level but also the possibility to compare demographic data between different nations’ registries. The Severe Asthma Network Italy (SANI) registry is promoted by allergy (SIAAIC) and respiratory (SIP/IRS) Italian scientific societies, GINA Italy and Federation of Italian Patient Associations (FEDERASMA), and has recruited severe asthmatics from more than 50 asthma centers [14]. The first descriptive analysis was published on 400 subjects [15]. In investigating the comorbidities, a high incidence of chronic rhinosinusitis with nasal polyps (CRSwNP) was found. This is important given the impact on quality of life (QoL), the response to biologics and the use of oral corticosteroids. The impact of CRSwNP is therefore prompting to investigate more properly this comorbidity in any severe asthma patient. Integrated data can provide important insights as in the Severe Heterogenous Asthma Registry Patient-centered (SHARP) and International Severe Asthma Registry (ISAR) registries, thus strongly supporting the creation of national registries. Such integration will provide a large body of new data. From registries on severe asthma, oral corticosteroid use has been declared in two-thirds of patients. This finding is consistent with other published reports, thus strongly highlighting the overuse of oral corticosteroids (OCS) in severe asthmatics. On the basis of this observation, SANI conducted a pharmacoeconomic analysis of the costs of OCS-induced side effects, and the findings stressed the high cost per year of this pharmacological intervention [16]. This is an example of how registries-derived information contributes to design strategies to reduce the high impact of corticosteroids in severe asthmatics [17].

### 2.5. The Perspective of the Medical Educator

Asthmatics looking for medical care invariably interact with more than one health professional. These figures and their collaboration increase with the complexity of the patient’s needs, which is high in the most severe forms of the disease. Commonly, physicians’ attitudes when coping with chronic diseases are rooted in the professionals’ experiences based on everyday general practice. There is therefore a demand of continuing medical education to improve the quality of the medical care and to ensure the best medical assistance. This educational process is founded on the delivery of high-quality, unbiased and standardized training in order to facilitate greater participation of health professionals in asthma management. National training centers at the primary or secondary level care should aim at providing the skills to efficiently deal with the requests of the severe asthmatic patients. Standards for asthma educators should be set in each country and supported by the scientific societies and the academic institution with structured preparation and carefully constructed and validated certification examinations. Unfortunately, a mismatch exists between the needs of the asthmatic patients and their families and the delivery of high-quality education to health professionals. Interesting observations related to the specific training proposed for current medical students come from the Italian experience. Despite epidemiological data showing an increase of morbidity, disability and mortality of respiratory disease all over the world, in Italy, postgraduate schools in respiratory diseases are found in only 25 out of 43 universities offering medical education, and the session on respiratory diseases during the academic program is not always held by a professor of respiratory medicine. In the residents’ program of Italian schools in respiratory medicine, almost 70% is reserved to the development of training activities [18]. When looking at the topic of asthma, clinical immunology and complete pulmonary functional tests are mandatory requirements for any specialization program in respiratory medicine. In this respect, the GINA documents [5,19] are excellent tools that should be integrated in the curriculum, given the rapidly evolving concepts in this field. It is obvious that the training course for schools of respiratory diseases should include the most updated knowledge in the underlying physiological mechanisms and comorbidities, for the overall management of the asthmatic disease, especially in its severe forms. It is logical to predict that guidelines are more likely to change practice when they are disseminated to clinicians using a specific educational intervention or are included in courses for university graduates.

## 3. What Makes Asthma a Severe Disease?

Severe asthma differs from the mildest (or less severe) forms of the disease in several respects. Certainly, severe asthma is characterized by a component of irreversible airflow obstruction and peripheral airways disease, with features of mainly neutrophilic inflammation, although in some subgroups it is associated with persistent eosinophilic inflammation, in a mixed type inflammatory phenotype. This suggests that severe asthma might be a different form of the disease, rather than mild asthma gone bad. Structural changes associated with persistent airflow limitation in asthma, also referred to as airway remodeling, include thickening of the airway wall due to subepithelial fibrosis, hypertrophy or stiffening of airway smooth muscle and hyperplasia of mucous glands and goblet cells. In addition, edema, vascular dilatation and increased numbers of blood vessels may contribute to the thickening of the airway wall. Inflammation per se has been suggested to be responsible for part of the remodeling process. Neutrophilic inflammation in general appears to be associated with the severity of asthma, although a subpopulation of severely asthmatic patients exhibits elevated numbers of eosinophils. In addition to non-reversible features of airway obstruction, the severe asthmatic phenotype is characterized by a condition of air trapping, as a result of early airway closure during expiration, which is likely to facilitate excessive airway narrowing, being therefore responsible for severe, and even fatal, asthma attacks. These pathological and functional alterations set the basis for the occurrence of several clinical patterns of severity, whose heterogeneity is influenced by endogenous or exogenous factors. The following sections highlight the key role of comorbidities, beta-2 agonist resistance and aging in complicating the course and the management of the disease. 

### 3.1. Comorbidities 

Comorbidities associated with severe asthma are common, complicate the overall management and influence patient outcomes. They can often interact, contributing to poor disease control and sometimes mimicking asthma symptoms. Several comorbidities are more common in severe asthma than in mild to moderate asthma or in healthy individuals. While some comorbidities, such as gastroesophageal reflux (GERD), bronchiectasis and COPD, are well recognized [20], many others remain unrecognized and detected only in an expert specialist setting. There are data indicating that the presence of comorbidity is associated with worse outcomes in patients with asthma. As an example, the presence of chronic rhinosinusitis is a strong risk factor for frequent exacerbations [21]. Comorbidities can be grouped into two large domains, the respiratory and the extra-respiratory ones [22]. Respiratory comorbidities can generally be classified according to their anatomical site along the respiratory tract: upper, middle and lower respiratory tract disorders. Most of them are usually known and therefore assessed during evaluation of the patient with severe asthma, while others such as dysfunctional breathing or vocal cord dysfunction [21,23], if not properly recognized, may lead to inappropriate escalation in asthma treatment. Extra-respiratory comorbidities are even more complex in the management of patients with severe asthma. It is therefore necessary to standardize a multidimensional approach in managing the patient with severe asthma, using screening questionnaires. The use of validated screening questionnaires for each comorbidity can increase their detection compared with a clinical consultation alone, guide the clinician to confirm the presence of the comorbidity and then address this problem with a multidisciplinary team. A recent meta-analysis has collected studies that systematically addressed comorbidities as part of their protocol, demonstrating that this approach leads to improvements in asthma control, quality of life and exacerbation rate [22].

### 3.2. Tolerance and Resistance to β2-Agonists 

Inhaled β2-agonists are indicated in the treatment of asthma. In the last century however, paradoxical asthma exacerbations and deaths have been associated with long-acting β2-agonists (LABAs), mostly when used without ICS. Preclinical and clinical studies documented that long-term use of LABAs was associated with increased effects of airway hyperresponsiveness (AHR) and loss of bronchoprotection [24]. In recent years, our knowledge on molecular pathways related to G protein-coupled receptors (GPRs) activation allowed us to understand the mechanism involved in desensitization and tolerance to β2-adrenergic receptors (β2-AR) [25]. The canonic mechanism of action of these drugs is the activation of β2-AR on airway smooth muscle leading to G protein activation and subsequent generation of c-AM: cAMP subsequently phosphorylates protein kinase A (PKA), involved in the control of airway smooth muscle tone, leading to reductions in intracellular calcium, smooth muscle relaxation and bronchodilation. However, there is now growing evidence that suggests that the binding of β2-agonists to β2-AR is pleiotropically coupled to many intracellular pathways, whereby, depending on the state of the β2-AR when activated, a subset of different intracellular responses can be triggered. Studies over the past 15 years have unveiled novel signaling pathways for β2-AR. Apart from the canonical mechanism of which activation involves the increase in cAMP signaling, another signaling pathway has been identified that involves activation of downstream mitogen-activated protein kinases (MAPK) like ERK1/2, JNK and/or p38 by arrestin. This is called biased agonism (or functional selectivity), and this type of activity has now been observed with different types of GPCRs, not just β2-AR. These different responses, that depend on what state the receptor is in when the ligand binds to it, can subsequently influence the intracellular signaling that in turn can influence the efficacy of β2-AR ligands. These findings explain why the long-term treatment with β2-agonists could be related to a loss of efficacy and tolerance induced by desensitization following repeated activation [26].

### 3.3. Aging 

It is clearly established that asthma in the elderly exists and is different from COPD under clinical and pathology aspects. However, it is less clear whether asthma in the elderly is more severe compared with younger individuals. Unfortunately, studies directly exploring severe asthma in the elderly are scanty. To overcome this, it could be of interest to explore whether aging is a risk factor for uncontrolled asthma. Studies on asthma control [27] demonstrated poorer control in the elderly compared with younger patients. However, the elderly may not perceive their symptoms, due to a blunted perception of dyspnea [28], and dyspnea may be misinterpreted or not reported, often considered as due to aging. In elderly patients, the presence of comorbidities and the low adherence to inhaled medication can negatively influence asthma control [29]. Indeed, the prevalence of chronic concomitant diseases is higher in the elderly and unintentional non-adherence is often present in this population. Cognitive impairment and lack of proper inhaled technique of respiratory drugs could often account for unintentional non-adherence in the elderly. A cluster analysis [30] on elderly asthmatic phenotypes demonstrated that patients with a long symptom duration and marked airway obstruction have a shorter amount of time to first acute asthma exacerbation. In the elderly, exacerbations are often serious: the age-adjusted mortality rate is higher in asthmatics aged 65 years and more. This observation could be due to the well-known issue of under-diagnosis and under-treatment of asthma in the elderly [31]. Furthermore, the blunted perception of dyspnea in the elderly could lead the patients to a severe airway obstruction without “alert signals” and therefore without the appropriate use of (rescue) medications. Finally, the elderly asthmatics are at risk of severe airflow limitation. It has been proposed that the combination of physiological age-related changes in the lung (the so-called senile lung) with the typical features of asthma in the elderly (e.g., slight differences in the inflammation patterns, medication adherence) may lead to more severe airway obstruction and, in turn, to more difficult-to-manage asthma in the elderly [32]. Asthmatics seem to lose some protective mechanisms with aging, such as the bronchodilator effect of deep inspiration [33], and they also show more severe airway obstruction compared with younger asthmatics [27]. In conclusion, several evidences show that older age could be a risk factor for difficult-to-control or even severe asthma.

## 4. What Are the Most Appropriate Tools to Assess/Monitor Severe Asthma?

### 4.1. Second Level Functional Assessment

Commonly employed spirometric tests, such as forced expiratory volume in one second (FEV_1_), forced vital capacity (FVC) and the ratio of FEV_1_ to forced vital capacity (FEV_1_/FVC), represent first-level functional assessments and are currently used in clinical practice to establish the degree of functional impairment and the magnitude of reversibility of bronchial obstruction. Measures of lung volumes, such as residual volume (RV) as well as total lung capacity (TLC), are, to some extent, correlated to the functional state of the peripheral airways. The importance of the peripheral airways in the pathological processes leading to severe asthma and the correlations with the severity of clinical manifestations and with patient-reported outcomes has led to the concept of severe asthma as a disease mainly of the small airways [34]. 

The single-breath nitrogen (sbN_2_) wash-out test is another non-invasive tool to detect early closure of peripheral airways. It is based on the detection of regional differences in ventilation distribution, represented by the phase III slope of the flow-volume curve following the wash-out test. Asthma patients who have frequent exacerbations (≥2 per year) have a higher degree of small airway disease measured with the single-breath nitrogen wash-out test than patients with infrequent exacerbations (<2 per year), whereas their FEV_1_ and FEV_1_/FVC are similar [35]. The multiple breath nitrogen wash-out (MBW) test measures functional residual capacity (FRC). The esophageal balloon technique represents the gold standard for the measurement of changes in pleural pressure, and is one of the most reliable methods to disclose the premature closure of small airways. Although a comprehensive lung functional assessment is advocated based on their clinical consequences in terms of identifiable therapeutic targets, this can be reasonably conducted only in a limited number of referral centers.

### 4.2. Imaging

Lung and airway imaging have the potential to improve our ability to quantify treatment success. There are several imaging modalities available for the lungs: chest radiograph (CXR), multi-detector computed tomography (MDCT), magnetic resonance (MR), positron emission tomography (PET), fluoroscopy, ultrasound (US) and optical coherence tomography (OCT). One important use of imaging modalities such as CXR and MDCT is to identify co-morbidities not related to asthma but which can nevertheless mimic or worsen asthma symptoms. These include pneumonia, pneumothorax, pneumomediastinum, allergic bronchopulmonary aspergillosis, chronic eosinophilic bronchitis and eosinophilic pneumonia and eosinophilic granulomatosis with polyangiitis for which specific treatment should be offered [36]. While modalities such as MDCT and MR have been used on a research basis to image changes in the lungs and airways specific to severe asthma, their usefulness to direct treatment is unclear. MR imaging using hyperpolarized noble gases such as 3He and 129Xe can be used to image ventilation inhomogeneity [37]. One can see areas of low and no ventilation in individuals with asthma at baseline and during acute exacerbations. However, it remains unclear how this information will improve our ability to treat severe asthma. CT scanning can be used to image air trapping and airway wall thickness [38]. With worsening disease, air trapping increases. This can now be visualized easily with 3-D reconstructed CT images. However, the same information can be acquired with less cost and no radiation exposure using body-plethysmography. While MDCT can visualize airway walls in vivo, MDCT is unable to identify any of the structures within the wall and can only visualize the wall as a solid structure. It is well established that the airway walls of individuals with asthma, especially the most severe kind, fatal asthma, are thicker on average than healthy individuals [39]. This has also been demonstrated in vivo using MDCT [40]. However, the difference in wall thickness in vivo is small and there is considerable overlap between healthy and diseased airways. Therefore, thicker airway walls based on MDCT images are unlikely to be helpful in identifying and treating those with severe asthma.

OCT is a newer imaging technology that can visualize the microstructure of biological structures in vivo. In addition, recent advances allow OCT to generate three-dimensional images with micrometer resolution. OCT does not use ionizing radiation and can generate images in real-time. OCT is the optical analog to B-mode ultrasound. However, instead of using sound waves, OCT uses low-coherence near-infrared laser light. OCT can also be deployed through a standard fiberoptic bronchoscope, making it easily deployable in clinical situations. The greatest strength of OCT is its ability to differentiate the various tissue layers in the airway wall [41]. The most important tissues in asthma are the epithelial layer, the basement membrane layer, the airway smooth muscle (ASM) layer and the glands. The use of OCT with bronchial thermoplasty (BT) therapy for severe asthma has several benefits. First, OCT may be useful in optimizing the location of maximum ASM for treatment. Second, OCT can verify the treatment success of BT (i.e., changes in the structural components of the airway wall post-BT). Third, OCT may be valuable in selecting the right patients for BT treatment. OCT and BT in combination may be an optimal way to treat severe asthma.

### 4.3. Biomarkers

The so called “SAVED” model has been suggested to outline the utility of a proposed biomarker: “Superior”— outperforms current practice; “Actionable”—having the potential to change patient management; “Valuable”—able to improve patient outcomes; “Economical”—to be cost-saving or cost-effective; “Deployable”—technologists should be in place for their assessment [42]. Figure 2 describes the steps required to move from a research setting to clinical applicability. The quest for biomarkers is linked up to the introduction into the treatment of asthma of formulations, many of which are biologic products targeting specific molecules driving the different asthma endotypes. Thus, (induced) sputum cytology provides evidence of eosinophilic, neutrophilic, mixed cellular, as well as low cell numbers (paucigranulocytic) profiles of airway inflammation in asthma [43]. The separate cellular profiles are orchestrated by a plethora of cytokines against the background of specific neural networks. Our understanding is most advanced about eosinophilic, high T-helper (Th) type 2 airway inflammation. It is identified by high blood and sputum eosinophils, high serum total and allergen-specific immunoglobulin (Ig) E, interleukin (IL) 5, 4 and 13 and other mediators upstream the inflammatory cascade like IL33, IL25 and thymic stromal lymphopoietin (TSLP). Biologic therapies and also smaller molecules have been tailored to block these molecules or their receptors. On the other end of the spectrum are the cases of non-eosinophilic asthma. Its pathogenesis is poorly understood, particularly for a neutrophil cellular profile of the airways, with corresponding cytokines from Th1 and Th17 cells, and the involvement of type 3 innate lymphoid cells, leading to activation of macrophages and release of neutrophil chemokines such as C-X-C motif chemokine ligand 8. In line with the SAVED model, a lot of hope is placed on biomarkers of the exhaled human breath [44]. Thus, fractional exhaled nitric oxide (FeNO) is already broadly assessed in clinical trials of formulations for patients with eosinophilic, high Th2 airway inflammation. New approaches for capturing exhaled biomarkers rely on analysis of breath condensate, but are mostly at an experimental stage. Alternatively, integral methods for non-invasive assessment and monitoring of airway inflammation and remodeling exist: exhaled breath temperature measurement and electronic sensing (e-sensing) technologies (electronic nose), which are currently under development and/or validation.

### 4.4. Expert Systems and Artificial Intelligence 

Artificial intelligence (AI) is the simulation of human intelligence processes by computer systems. The expert systems (ES) are one component of AI. They are consulted to obtain advice, suggestions and recommendations on issues that fall within the human experts’ knowledge and are widely used in many different fields of activities (among which medicine). Typically, an ES incorporates: a knowledge base containing accumulated experience, and a rules engine (a set of rules for applying the knowledge base to each particular situation that is described to the program); the system’s capabilities can be enhanced with additions to the knowledge base or to the set of rules. In brief, an ES is a computer program that uses AI technologies to simulate the judgment and behavior of a human that has expert knowledge and experience in a particular field. Current systems may include machine-learning capabilities that allow them to improve their performance based on experience, just as humans do. Recently, a panel of Italian pulmonologists developed an ES for assisting the identification of individuals suffering from chronic obstructive lung disease (COLD) in primary care settings [45]. By removing diagnoses that cannot be made based on only the questionnaire, the overall accuracy of the COLD ES was 97.50%. This seems promising for a future use of the ES in different settings, including primary care, and raises the possibility that a similar effort can be made for the diagnosis of severe asthma. 

## 5. The Masked Facades of Severe Asthma

### 5.1. COPD and ACO 

It is easy to differentiate pure asthma from pure COPD because they reflect the extremes of a spectrum, but it is widely recognized that there are patients, especially those who are elderly, who present with features of both asthma and COPD, leading to an overlap [46]. The presentations of these illnesses can converge and mimic each other, making it difficult to give these patients a diagnosis of either condition also because some of the mechanisms driving airway obstruction and hyperresponsiveness are similar in asthma and COPD, and some are different. In any case, the association of asthma and COPD in the same patient has been designated mixed phenotype asthma-COPD or asthma–COPD overlap (ACO). However, a precise definition of ACO is lacking and this makes it difficult to categorize the contrasting distinctive features of this phenotype. Furthermore, ACO presents as multiple phenotypes, some including more features of asthma than COPD and the opposite in others [47], such as patients with COPD and eosinophilic inflammation, patients with asthma and severe disease or who smoke, in whom there is predominantly neutrophilic inflammation, and patients with asthma who have largely irreversible airway obstruction due to structural changes. Consequently, it has been suggested that ACO includes two main conditions, smoking asthmatics and eosinophilic COPD patients (≥300 blood eosinophils/μL), with different medication requirements and prognosis that should not be pooled together. Use of ≥300 blood eosinophils/μL as a treatable trait should be recommended [48]. The eosinophilic COPD should identify those COPD patients that would benefit the most from ICS. However, studies have not shown differences in the response to ICS therapy based on eosinophil counts, particularly in those with mild-to-moderate COPD [49,50]. Nonetheless, because of risk in patients with asthma with LABA monotherapy, it has been suggested that ICS (usually associated with a LABA) is the preferred therapy in ACO. The term ACO may be a relatively simple way to avoid excessive diagnostic investigations and streamline therapy. However, due to the absence of a clear definition and the inclusion of patients with different characteristics under this umbrella term, it may not facilitate treatment decisions, especially in the absence of clinical trials addressing this heterogenic population. Therefore, there is an open debate on whether the term “ACO” should be abandoned because it does not identify a clearly independent disease entity. This heterogeneity indicates a wide range of disease mechanisms [51]. Asthma and COPD are not only heterogeneous diseases but also associated with complex medical conditions. Different molecular characteristics associated with different endotypes may present in varying proportions in any given patient.

### 5.2. Eosinophilic Disorders

Among clinical conditions related to a pathogenic role of eosinophils, hypereosinophilic syndromes (HES), defined as conditions in which there is an elevation in the peripheral blood absolute eosinophil count greater than 1500/mL on at least two separate detections, should be included. HES can be divided in primary and secondary forms and, among them, eosinophilic granulomatosis with polyangiitis (EGPA), which is closely associated with asthma [52]. According to the American College Rheumatology classification, the diagnosis of EGPA can be made when at least four criteria are present [53]. As mentioned, asthma is a peculiar manifestation of EGPA, often presenting several years before EGPA, starting in association with CRSwNP. EGPA is classically considered a Th2-driven inflammatory response. Peripheral T cell lines from patients with EGPA produce large amounts of IL-4, IL-5 and IL-13. Elevated IL-5 is found in serum and bronchoalveolar lavage fluid of patients with active disease [54]. The potential role of Th17 and B cells, producing anti-neutrophil cytoplasmic antibodies (ANCA), in the pathogenesis of EGPA has been suggested [55]. A higher rate of Th17 cells able to secret larger amounts of IL-17A is found during active disease. Similarly, a reduced proportion of regulatory T cells, which have a suppressive effect on Th17 cells, and aberrant regulatory T cell function have been implicated. Despite a correct treatment with high doses of systemic corticosteroids, a significant proportion of patient relapses and immunosuppression is needed. In addition, the majority of patients do not achieve a good control of asthma symptoms despite inhaled and systemic corticosteroids. A clinical trial performed in EGPA patients with mepolizumab (300mg/4weeks) showed that, in comparison with the placebo, the biologic is able of inducing a remission of the disease in a significant proportion of patients even if after interruption of treatment they relapse [56].

Although not a frequent comorbid condition, eosinophilic esophagitis (EoE) carries some very interesting inflammatory and immune characteristics that are worth discussing as they include both similarities and differences from asthma. EoE is a chronic atopic clinical-pathologic disease defined by eosinophil infiltration limited to the esophageal epithelium [57,58]. Once considered a rare disease, it has now reached a yearly incidence of 1–2/1000 in the US with a 50–70-fold increase in the last decade [59]. It presents with symptoms of esophageal dysfunction such as dysphagia, food impaction, esophageal strictures, reflux-like symptoms, vomiting but also post-prandial cough and chest pain that can mimic asthma. Currently, diagnosis and follow-up of treatment of EoE are based on the count of eosinophils per high power field (eos/hpf) in esophageal biopsy obtained via esophagogastroduodenal endoscopy (OGD). EoE has many similarities with asthma in terms of inflammatory process and pathogenesis; however, the recent clinical trial with biologics as treatment of EoE points out some differences with asthma. EoE appears to be part of the atopic march and to have Th2-predominant inflammation driven by chronic antigen exposure. However, food but not environmental allergens seem to be significant triggers of EoE [60]. The complex nature of the immunological response constitutes a major obstacle in the development of targeted treatments. Eosinophils are very useful for diagnosis and monitoring of the disease but are not essential in EoE development. Indeed, anti-Il-5 monoclonal antibodies (i.e., reslizumab or mepolizumab), which are effective in blocking IL-5, have been ineffective in treating symptoms and inflammation in EoE in both children and adults [61,62]. Even if EoE patients are atopic and often produce IgE against environmental and food allergens, IgE does not appear essential to trigger or maintain EoE inflammation. Indeed, classical food allergen testing such as measurement of specific food IgE by in vivo (i.e., skin prick test) or in vitro testing (i.e., ImmunoCap, ISAC), has been unable to predict EoE food allergen triggers. In an animal model, EoE could still develop in the absence of IgE, and children who outgrow IgE-mediated food allergies are at risk of developing EoE. Therefore, it is not a surprise that omalizumab has been ineffective in treating symptoms and inflammation in EoE [63]. Anti-IL-4/IL-13 antibodies such as dupilumab with broader efficacy against Th2 inflammation have been shown to be effective in reducing inflammation and dysphagia in a small double-blind study placebo-controlled study [64]. Anti-IL-13 antibodies have shown efficacy against inflammation but not dysphagia [65]. In conclusion, EoE is an emerging comorbidity of asthma, and its presence may lead to use personalized medicine and therapies that can treat EoE and asthma at the same time.

## 6. Challenges in the Treatment of Severe Asthma

### 6.1. Current Algorithm (as Proposed by GINA)

The Global Initiative for Asthma, launched 25 years ago, produces a yearly updated clinically oriented, evidence-based strategy to help the translation of evidence into clinical practice, to promote optimal care of this disease around the world [6]. Among the various documents produced by GINA, summaries of its recommendations, related to current knowledge obtained from published research and guidelines, we find “Pocket Guides”, including a recent one on Diagnosis and Management of Difficult-to-treat and Severe Asthma in Adolescent and Adult Patients, first published in November 2018 and updated in April 2019 [19]. This booklet is intended for use by general practitioners, lung specialists and other health professionals involved in the management of patients with asthma. The recommendations in this Pocket Guide are based on evidence from good-quality systematic reviews or randomized controlled trials or, lacking these, good observational data, and on consensus by expert clinicians and researchers, when evidence is not available. The booklet includes a decision tree produced with the collaboration of experts in human-centered design, to enhance the usefulness of this document for end-users. The Pocket Guide should be ideally used in conjunction with the full GINA 2019 report. The prevalence of severe asthma will vary by country according to access to medications and the health system. In one study, 17% of asthma patients were considered to have “difficult-to-treat asthma” characterized by poor symptom control and/or exacerbations despite GINA Step 4 or 5 treatment. This was usually due to sub-optimal management and/or patient behavior, with only about 4% of patients with asthma having “truly” severe asthma, having still poor symptom control despite optimal management, good inhaler technique and good adherence to therapy [66]. The Pocket Guide also emphasizes that severe asthma is heterogeneous, with different characteristics (phenotypes) and different mechanisms of development or persistence (endotypes). Recommendations are therefore made for asthma related to patients with features of either Type 2 or non-Type 2 inflammation. The decision tree changed previous flowcharts and text-based information to a more detailed visual format. Specific identification of items related to diagnosis, decision points and treatment, in addition to locus of care considered and reminders about ongoing issues are clearly indicated. The first four sections of the decision tree are for use in primary care and/or specialist, the next three sections are mainly relevant to respiratory specialists and the last one is about collaborative care between the patient, the general practitioner, specialist and other health professionals. The document stresses the need to proceed with a systematic investigation of difficult-to-treat asthma, to distinguish asthma management problems from truly severe asthma and also provides relevant information on new biologics.

### 6.2. Drugs for COPD (Do They Also Work in Severe Asthma?)

The pharmacological classes used in asthma and in COPD are pretty similar. The differences are mainly in the strategy of the use of the medications. Traditionally, in obstructive lung diseases, inhaled and oral treatments have been firstly tested in asthma and then evaluated for COPD. The exception being triple therapy with ICS/LABA/long-acting muscarinic antagonists (LAMA) that has been widely used for a long time (as open triple with more than one device) in COPD before being tested in severe asthma [67]. Even the use of triple therapy in one single device has been developed in COPD before being tested in severe asthma. At the moment, we still do not have the full results of the trials comparing triple therapy in severe asthma. What we know from the PrimoTina study [67] is that the addition of tiotropium to the high dose ICS/LABA combination budesonide/formoteterol improves lung function and reduces exacerbations in a selected population with fixed airflow limitation, i.e., a very specific subgroup of severe asthma, those more similar to COPD. Conversely, triple therapy in COPD is effective in reducing exacerbation risk compared with ICS/LABA and LABA/LAMA combinations: in the latter case, the higher the blood eosinophils counts, the greater the magnitude of the superiority [68]. Whether it is the same in severe asthma has still to be evaluated. The level of eosinophils is (one of) the more popular biomarker used in severe asthma to assess the eligibility for biologic treatments. The same approach has been tested in COPD, following the same concept on the characterization of subjects: high blood eosinophils plus frequent exacerbations. With mepolizumab, the magnitude of the effect was lower than expected but the study clearly showed that the efficacy of mepolizumab is clearly selective on the prevention of exacerbations requiring systemic corticosteroids, but was ineffective in reducing exacerbations requiring antibiotic treatment. Though not surprising, this evidence identifies the population (phenotype) where the efficacy of an anti-IL5 should be expected: those patients with exacerbations not requiring antibiotics that more likely have eosinophilic exacerbations [69]. The results with anti-IL5Ra in COPD are overall less convincing [70]. However, subgroup analyses are being conducted to identify the population that could benefit from anti-IL5Ra treatment. In severe asthma and in severe COPD, clinical trials have tested the efficacy of macrolides (in particular azithromycin, with different strategies/dosages) as an option for patients not responding to other treatments, particularly for the prevention of exacerbations. In both clinical conditions, the efficacy of this approach has been documented, at the risk of development of antibiotics resistance. Given the WHO warning on appropriate use of antibiotics as resistance is becoming a global threatening problem, the chronic use of azithromycin should be limited to a highly selected population and should be the last resort in severe asthma or COPD patients where any other treatment has failed. A number of new molecules are under investigation both in severe asthma and COPD. Likely, we will see a cross-over of the evaluations moving from asthma to COPD or, occasionally, vice versa. Nothing new under the sky.

### 6.3. Vitamin D

A significant body of epidemiological evidence suggests associations between vitamin D insufficiency and worse asthma control. However, clinical trials of vitamin D supplementation in asthma, with varied design and dosing regimens, have produced variable outcomes. Most recently, meta-analyses and systematic reviews indicate that vitamin D reduced asthma exacerbations, had steroid-sparing effects and protected against acute respiratory tract infection. These effects were greatest in subjects who were very vitamin D-deficient and most effective when given frequently, i.e., daily/weekly, but not as a bolus [71,72]. Vitamin D beneficially modulates diverse immunological pathways linked to heterogeneous asthma endotypes. Allergic asthma is frequently characterized by the failure of tolerance and the development of pathologic responses to inhaled aeroallergens. Vitamin D may improve control of allergic asthma through demonstrated actions on dendritic cells to promote a tolerogenic phenotype, to enhance the frequency of inhibitory molecule expression (e.g., IL-10, CTLA4, Foxp3) by T regulatory cells (Treg), to reduce class switching to IgE and increase IL-10 synthesis in B lymphocytes and to decrease mast cell activation. In non-allergic eosinophilic asthmatic inflammation, vitamin D may act on bronchial epithelial cells to enhance antimicrobial and anti-inflammatory functions, block the activity of the alarmin IL-33 through stimulation of a soluble receptor antagonist, sST2, as well as dampen ILC2 and eosinophil activation. In glucocorticoid or treatment refractory asthma, vitamin D may act on bronchial epithelial cells to promote antimicrobial and anti-oxidant actions, to suppress Th17-associated cytokines, enhance anti-inflammatory IL-10 synthesis and on neutrophils to promote antimicrobial activity and reduce pro-inflammatory cytokine production. Vitamin D is a potent inducer of antimicrobial mechanisms in many cell types, which is of relevance in the control of infection-precipitated asthma exacerbations [73]. These mechanisms support a role for vitamin D in secondary prevention to reduce exacerbations and inflammation in asthma. These data indicate the potential that restoring vitamin D sufficiency can incrementally improve asthma control in existing disease. The most recently emerging studies suggested a role for vitamin D in the primary prevention of asthma. Vitamin D deficiency is often profound in pregnancy [74]. Offspring of women with high vitamin D levels in the first trimester that were sustained by supplementation during pregnancy had a significant reduction in recurrent wheeze and asthma at age 3 years [74,75].

### 6.4. Current Treatment

In an apparently uncontrolled asthmatic patient under standard inhaled therapy, the GINA document [6] suggests the following actions: a) check for adherence and evaluate the inhaler technique (in several cases, adherence to regular treatment is poor, inhaler technique is incorrect and when these aspects are appropriately corrected, asthma may result under control); new “intelligent” devices checking inhaler technique and facilitating drug inhalation may improve adherence; b) assess and appropriately treat asthma-associated comorbidities as these may be responsible for lack of control and their appropriate management may result in a better management of the disease; c) optimize current pharmacologic treatment using high-dose (and sometimes supramaximal doses) of ICS associated with LABA and frequently with tiotropium or montelukast. New high-strength ICS/LABA combinations, like BDP200/formoterol or fluticasone furoate/vilanterol 184/22, have demonstrated greater efficacy in improving symptoms and lung function in comparison with lower doses of the same combinations [76,77]. Furthermore, the SMART strategy in patients already treated with a regular high-dose budesonide/formoterol combination was able to reduce the exacerbation rate better than other options [78]. In severe uncontrolled asthmatics, the addition of tiotropium to high-dose ICS/LABA resulted in a significant improvement in FEV_1_ and a mild but significant reduction in severe exacerbations [67]. Finally, in a real-life open study, the addition of montelukast to ICS/LABA combinations improved asthma control over time [79]. Despite any optimization of the current therapy, a large portion of severe asthmatics remains uncontrolled. In these patients, GINA guidelines recommend to proceed to pheno-endotyping for the identification of type 2 asthma, for which old (anti-IgE) and new (anti-IL5 or anti-IL5R, or anti-IL4/13R) biologics are effective, as demonstrated in phase III trials where these drugs reduce by about 50% the rate of severe exacerbations in comparison with placebos, and in oral steroid-sparing studies. Clinical indications are similar for all these biologics (severe uncontrolled asthma despite high level of current therapy), but the prescriptive criteria are different. Anti-IgE treatment (omalizumab) has been shown to be effective in early onset severe allergic asthma, often associated with allergic rhinitis or chronic urticaria, with definite levels of serum IgE plus allergic sensitivity to perennial allergens. A large amount of real-life observational studies have demonstrated that omalizumab is effective independently from the level of blood eosinophilia [80], although its efficacy seems lower in older patients and in patients with persistent eosinophilia [81]. Efficacy of anti-IL5 or anti-IL5R (mepolizumab or benralizumab) has also been shown in late-onset severe asthma [82], often non-allergic and with nasal polyps, with blood eosinophilia (cut-off values are slightly different for mepolizumab and benralizumab). Blood eosinophil levels, the rate of severe exacerbations and the presence of nasal polyps are good predictors of response to anti-IL5 or anti-IL5R [83]. Treatment with anti-IL4/13R (dupilumab) has been demonstrated to be effective in late-onset asthma, often allergic and with nasal polyps or atopic dermatitis (on which dupilumab is strongly effective) [84], associated with mild blood eosinophilia (>150 cell/μL) and/or increased levels of exhaled nitric oxide (>25 ppb). Therefore, considering the similar clinical and biological profile for the prescription of these biologics, the presence of comorbidities on which the same biologics are effective may guide the choice of the preferred treatment. Unfortunately, up to 50% of severe uncontrolled asthmatics are not affected by type 2 asthma: in these patients, new targets for effective treatment are needed.

### 6.5. Future Directions 

It is increasingly recognized that severe asthma comprises several distinct phenotypes or endotypes that may require specific therapies and that precision medicine will be increasingly used more effectively in managing these patients. Figure 3 describes the main therapeutic targets in severe asthma. Approximately 50% of patients with severe asthma have a type-2 immunity (T2) pattern of inflammation, with increased eosinophils, and these patients usually respond well to antibodies against IL-5 (if blood eosinophils are high and there are frequent exacerbations) or anti-IL-4Rα (if exhaled nitric oxide (FeNO) is high and especially if they suffer from concomitant atopic dermatitis and rhinosinusitis) [85]. Tezepelumab, which targets TSLP, may be even more effective as it is upstream of both IL-4/13 and IL-5, but may also target non-T2 mechanisms [86]. Anti-IL-33 (such as etokimab) and anti-IL-33 receptor (ST2) may also have a similar broader efficacy. A much greater challenge is the development of more precise therapies for non-T2 severe asthma, which includes neutrophilic asthma and paucigranulocytic asthma. So far, targeting neutrophilic asthma has included unsuccessful efforts with CXCR2 antagonists, which inhibit the recruitment of neutrophils into the airways by blocking the effects of CXCL8 and related chemokines [87,88]. Similarly, blocking IL-17 receptors with brodalumab was also ineffective, although patients with severe neutrophilic asthma were not selected [89]. There is evidence that NLRP3 inflammasome activation may be important in neutrophilic severe asthma, including asthma associated with obesity, and associated with increased IL-1 and caspase-1 [90]. An inhibitor of NLRP3 is effective in a mouse model of severe neutrophilic asthma [91]. A novel endotype of severe asthma has been recognized from the U-BIOPRED cohort of severe asthma that involves IL-6 receptor trans-signaling that may respond specifically to an IL-6 receptor antibody, such as tocilizumab [92]. Novel broad-spectrum anti-inflammatory treatments, such as phosphodiesterase-4, JAK and IRAK-4 inhibitors may be useful in non-T2 inflammation [93,94]. Macrolide antibiotics are also effective in reducing exacerbations of non-T2 asthma [95,96]. The potential for antibiotic resistance is leading to a search for non-antibiotic macrolides that retain anti-inflammatory effects. For paucigranulocytic patients, anti-inflammatory treatments are not indicated, but they may benefit from adding long-acting muscarinic antagonists [67]; highly selected patients may benefit from bronchial thermoplasty and targeted lung denervation.

### 6.6. Non Pharmacological Treatments

The most recent edition of the GINA guidelines [5] includes bronchial thermoplasty (BT) as a non-pharmacological, selective treatment of severe asthma [97]. BT delivers controlled thermal energy to the airway wall through a dedicated catheter during a series of bronchoscopy procedures that result in a prolonged reduction in ASM mass, thus ameliorating symptoms of asthma. Since the first randomized controlled trial of BT in asthma, the Asthma Intervention Research (AIR) study’s [98], which showed improvements in asthma-related quality of life and reduction in the frequency of exacerbations following treatment with BT, large randomized clinical trials with longer-term follow-up have supported the beneficial effect of the procedure [99,100], which has been then confirmed in real-life settings [101]. Many unanswered questions with regard to long-term maintenance of efficacy and need for expertise require that the procedure is used in selected patients, in dedicated centers. 

Pulmonary rehabilitation (PR) is a comprehensive intervention which includes exercise training, education and behavior change, designed to improve the physical and psychological condition of people with chronic respiratory disease and to promote the long-term adherence of health-enhancing behaviors [102]. The main goal is to impact dyspnea/fatigue and chronic respiratory symptoms. A recent study showed that in severe asthmatics, inspiratory capacity (IC) significantly drops during the 6MWT to the same extent as COPD subjects with a similar degree of lung impairment, indicating the development of dynamic hyperinflation [103]. Contrary to COPD, the occurrence of dynamic hyperinflation in asthmatic subjects does not seem to be associated with changes in dyspnea perception. These differences should be taken into account in rehabilitation settings to personalize treatment. A home-based PR program can be offered to many patients and can significantly improve all respiratory function parameters [104]. Completion of PR by a population with asthma results in improvements in exercise tolerance, weight loss, depression and quality of life [105]. Depression is a risk factor for non-implementation of PR in patients with asthma. Improvement in asthma control is greater in patients with uncontrolled asthma than in patients with partially controlled asthma after PR [106]. Therefore, patients with uncontrolled asthma should be given opportunities to benefit from PR programs.

Lung transplantation (LT) can be considered another non-pharmacological approach to severe asthmatic patients. Potential candidates are patients with fixed obstruction secondary to remodeling; however, recent biological therapies have changed the natural history of functional impairment. In fact, the recent literature on the topic is poor and consists of case reports or limited asthma patient series [107,108,109]. Interestingly, it has been reported that non-asthmatic recipients of asthmatic lungs may develop asthma after transplantation. On the other hand, the asthmatic recipients of normal lungs did not develop asthma up to three years after transplantation. Selection criteria as well as contraindications can be assimilated to those for COPD patients.

## 7. Severe Asthma in Acute or Difficult to Treat Settings

### 7.1. Acute Settings 

Patients at increased risk of asthma-related death should be identified to prevent a flare-up (or exacerbation). Risk factors include hospitalization or emergency care for asthma in the last 12 months, any history of near-fatal asthma requiring intubation and ventilation, not currently using ICS, or poor adherence with ICS, currently using or recently stopped using OCS, over-use of SABAs (especially if more than one canister/month), lack of a written asthma action plan, history of psychiatric disease or psychosocial problems, and confirmed food allergy. The current optimal pharmacological management of asthma flare-ups is simple and straightforward, including SABA, inhaled ipratroprium bromide, oxygen supplementation to maintain SpO2 between 93% and 95% in adults and 94% and 98% in children, oral and IV corticosteroids, and IV magnesium. Indeed, other pharmacological approaches sometimes used in the clinical practice [110,111,112] did not show to be more effective than the standard treatment. The same was apparent for intravenous aminophylline which did not demonstrate any additional bronchodilation compared with standard care with beta-agonists with significant adverse effects [113]. Moreover, the existing evidence does not provide support for the administration of helium–oxygen mixtures to all emergency department patients with asthma flare-up, with some evidence of certain beneficial effects in patients with more severe obstruction [114]. Very limited and sometimes anecdotic data are present for general anesthesia drugs, or propofol and ketamine [115]. Finally, the recent ERS document on NIV, given the uncertainty of evidence, was unable to offer a recommendation of the use of NIV for acute respiratory failure due to asthma [116].

### 7.2. Pregnancy

Asthma is the most common chronic medical condition reported during pregnancy and its prevalence in the population has increased in recent decades, representing a significant public health issue. Maternal asthma is associated with an increased risk of adverse perinatal outcomes [117] such as pre-term birth, low birth weight, small for gestational age infants, perinatal mortality and pre-eclampsia. Studies have reported a relationship between increased asthma severity, decreased asthma control or asthma exacerbation and increased perinatal complications [118]. Due to several mechanisms (i.e., maternal hormones, beta-adrenoreceptor responsiveness, fetal sex, altered immune function, physiological respiratory changes), most women with asthma experience an unpredictable change in asthma control while pregnant. Up to 45% of pregnant women with asthma have moderate or severe exacerbations requiring medical intervention during pregnancy and approximately 6% of them are hospitalized with a severe asthma flare-up [119]. The risk of asthma exacerbation during pregnancy is a function of asthma severity [120], and correlates with non-adherence to ICS. In fact, despite data indicating the safety of asthma medications during pregnancy, asthma remains undertreated in pregnant women and at least one-third of patients are non-adherent to inhaled therapy. In addition, changes in the maternal cell-mediated immunity during pregnancy may make pregnant women more susceptible to viral infections, leading to exacerbations. International guidelines provide the recommendations for successful clinical management of asthma in pregnancy. Optimizing asthma management in pregnancy is pivotal to protect the health of both mother and fetus. Thus, maintaining fetal oxygenation by preventing maternal hypoxia is the goal of treatment. With this aim, GINA recommendations highlight the importance to aggressively treat acute exacerbations during pregnancy with SABA, oxygen supplementation and early administration of systemic corticosteroids. Preconception care, regular review of asthma every 4–6 weeks, stepwise approach, identification and management of comorbidities are also recommended.

There is good evidence regarding the safety of the major drugs classes used to treat asthma during pregnancy, including ICS, especially for budesonide, and β2-agonists. A UK population-based study [121] evaluated the effect of maternal use on congenital malformations. Authors found that gestational exposure to commonly used asthma medications was safe overall, although a teratogenic risk of cromones could not be excluded. The Xolair Pregnancy Registry (EXPECT) evaluated the safety of omalizumab in pregnant severe asthmatic women. Data collected demonstrated that the prevalence of major congenital defects in EXPECT was not higher than those reported in the general population with asthma [122]. In addition, omalizumab does not increase the risk of pre-term birth or small for gestational age infants.

## 8. Conclusions

Severe asthma may be frustrating to treat. Patients with severe unremitting disease have the greatest impairment of quality of life and account for a disproportionate use of health-care resources through hospital admissions, unscheduled doctor visits and use of emergency services, thereby accounting for a significant medical and societal burden. This is because gaps still exist in the knowledge of the exact mechanisms underlying this condition. These proceedings of the International Course on Severe Asthma summarize the main topics that were discussed with the aim to highlight the many questions that remain, which, if adequately addressed, may eventually lead to improved outcomes. Severe asthma is not a rare entity and is no longer an orphan disease. Understanding whether severe asthma is the aggressive phenotype of a mild disease gone bad or a different disease with particular endotypes may also help properly identify the targetable and treatable features of all asthmas.

## Figures and Tables

**Figure 1 jcm-09-02397-f001:**
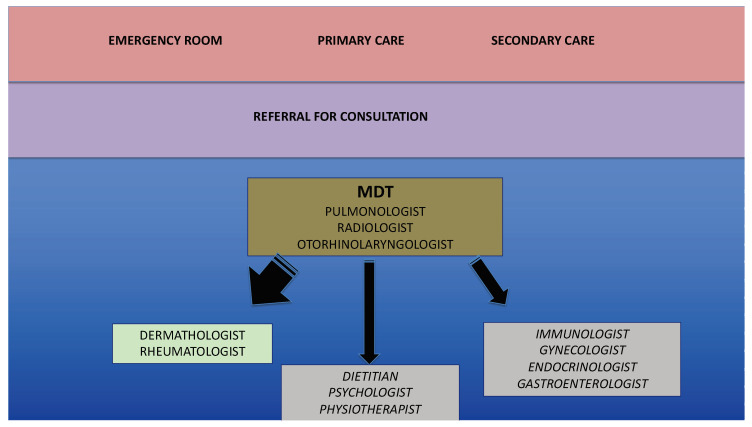
Staff and specialties involved in a severe asthma service with a multidisciplinary approach. The dimensions of the arrows indicate the frequency of consultations.

**Figure 2 jcm-09-02397-f002:**
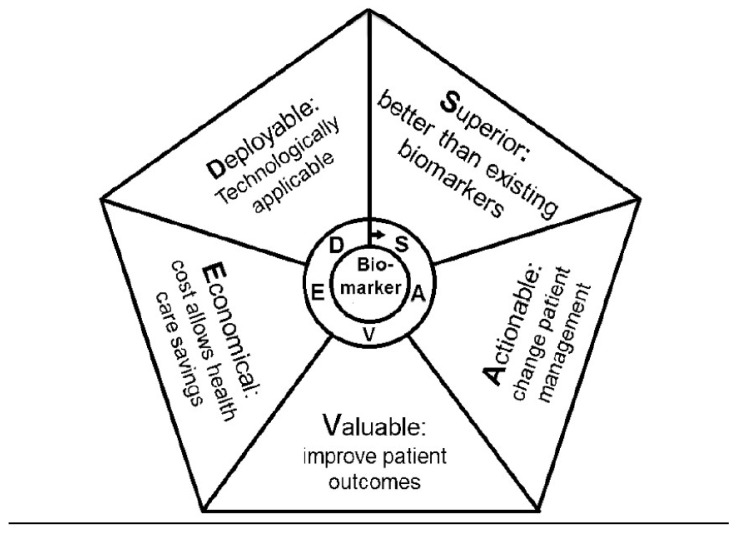
The “SAVED” approach to characterize an “ideal” biomarker. See text for details.

**Figure 3 jcm-09-02397-f003:**
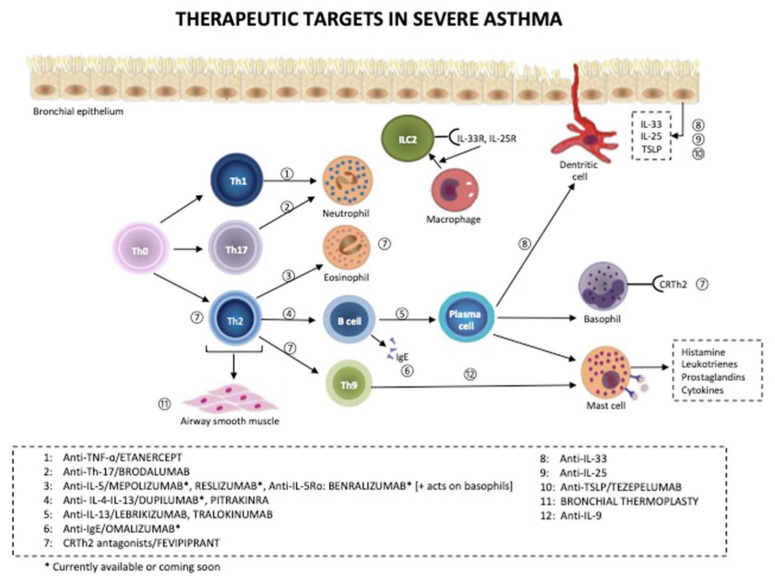
Inflammatory pathways of severe asthma and potential targets of biologic drugs.

**Table 1 jcm-09-02397-t001:** Lists of topics identified from the Delphi consensus.

Unanswered Questions	Specific Topics
**What are the different faces of severe asthma?**	The patient’s perspective
	The clinician’s perspective
	The researcher perspective
	The scientific society’s perspective
	The academic’s perspective
**What makes asthma a severe disease?**	Comorbidities
	Tolerance and resistance to β2-agonists
	Aging
**What are the most appropriate tools to assess/monitor severe asthma?**	Second level functional assessment
	Imaging
	Biomarkers
	Expert systems and artificial intelligence
**Has severe asthma masked facades?**	COPD and ACO
	Eosinophilic disorders
**What are the challenges in treating severe asthma?**	Current algorithm
	Drugs for COPD
	Vitamin D
	*Future directions*
	*Biologic drugs*
	*Non pharmacological treatments*
**Severe asthma in acute or difficult settings**	*Acute settings*
	*Difficult settings (pregnancy)*

**Table 2 jcm-09-02397-t002:** Evolution of the definition of severe asthma.

Year	Definition	Endorsement
1999 (Chung K.F., Eur. Respir. J.) [1]	difficult/therapy resistant asthma	ERS
2000 (ATS workshop, Am. J. Respir. Crit. Care Med.) [2]	refractory asthma	ATS
2007 (Chanez P., J. Allergy. Clin. Immunol.) [3]	severe asthma	GINA
2014 (Chung K.F., Eur. Respir. J.) [4]	severe controlled and uncontrolled asthma	ERS/ATS
2018 (GINA pocket guide) [5]	difficult-to-treat asthma severe asthma	GINA

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
