# Peer review of "The Hidden Burden of Severe Asthma: From Patient Perspective to New Opportunities for Clinicians"

_jcm, 2020, doi:10.3390/jcm9082397_

Round 1

Reviewer 1 Report

Comments to the authors:

This paper is well written and provides useful information on clinical practice of severe asthma. However, there are some concerns to be clarified.

Line 284, …with features of mainly neutrophilic inflammation: Please refer to the mixed type inflammation involving both eosinophils and neutrophils.

Line 357: Please check if the sentence is correct.

Line 393, ….they poorly correlated to the severity of the disease: Oscillometry, also known as the FOT, in severe asthma is described very shortly. These should be deleted so as not to confuse the readers. Otherwise, the authors should comment thoroughly with references

Line 609: Please check if the sentence is correct.

Author Response

This paper is well written and provides useful information on clinical practice of severe asthma. However, there are some concerns to be clarified.

Line 284, …with features of mainly neutrophilic inflammation: Please refer to the mixed type inflammation involving both eosinophils and neutrophils.

The Reviewer is right. We have changed the sentence accordingly.

Line 357: Please check if the sentence is correct.

The sentence has been changed as correctly pointed out.

Line 393, ….they poorly correlated to the severity of the disease: Oscillometry, also known as the FOT, in severe asthma is described very shortly. These should be deleted so as not to confuse the readers. Otherwise, the authors should comment thoroughly with references

 We have followed the suggestion of the Reviewer and removed the sentence that refers to oscillometry and FOT to avoid confusion for the reader.

Reviewer 2 Report

It is a very nice review article concering severe asthma perspectives. I have short comments on that.

  1. Line 336, 2AR->beta2AR
  2. Line 739, it would be better if the authors could add more current outcomes of BT.

Author Response

Line 336, 2AR->beta2AR

The thank the Reviewer for pointing this out. Appropriate change has been made.

Line 739, it would be better if the authors could add more current outcomes of BT.

We have expanded the paragraph has properly requested.